

# Evaluation of WRF-DART (ARW v.3.9.1.1 and DART manhattan release) multi-phase cloud water path assimilation for short-term solar irradiance forecasting in a tropical environment

Frederik Kurzrock[1,2], Hannah Nguyen[2], Jerome Sauer[2], Fabrice Chane Ming[3], Sylvain Cros[2], William L. Smith, Jr.[4], Patrick Minnis[4], Rabindra Palikonda[4], Thomas A. Jones[5], Caroline Lallemand[2], Laurent Linguet[6], and Gilles Lajoie[1]

[1]Institut de Recherche pour le Développement (IRD), UMR 228 ESPACE-DEV, Université de La Réunion, Saint-Denis, La Réunion, France
[2]Reuniwatt SAS, Sainte Clotilde, La Réunion, France
[3]Laboratoire de l'Atmosphère et des Cyclones, UMR8105, UMR CNRS - Météo-France - Université, Université de La Réunion, La Réunion, France
[4]Climate Science Branch (E302), NASA Langley Research Center, Hampton, Virginia, USA
[5]Cooperative Institute for Mesoscale Meteorological Studies, University of Oklahoma, Norman, Oklahoma, USA
[6]Institut de Recherche pour le Développement (IRD), UMR 228 ESPACE-DEV, Université de Guyane, Cayenne, Guyane, France

**Correspondence:** Frederik Kurzrock (frederik.kurzrock@reuniwatt.com)

**Abstract.** Numerical weather prediction models tend to underestimate cloud presence and therefore often overestimate global horizontal irradiance (GHI). The assimilation of cloud water path (CWP) retrievals from geostationary satellites using an ensemble Kalman filter (EnKF) led to improved short-term GHI forecasts of the Weather Research and Forecasting (WRF) model in mid-latitudes in case studies. An evaluation of the method under tropical conditions and a quantification of this

5 improvement for study periods of more than a few days is still missing. This paper focuses on the assimilation of CWP retrievals in three phases (ice, supercooled, and liquid) in a 6-hourly cycling procedure, and on the impact of this method on short-term forecasts of GHI for Reunion Island, a tropical island in the South-West Indian Ocean. The multi-layer gridded cloud properties of NASA Langley's Satellite ClOud and Radiation Property retrieval System (SatCORPS) are assimilated using the EnKF of the Data Assimilation Research Testbed (DART) manhattan release (revision 12002) and the advanced research WRF

10 (ARW) v3.9.1.1. The ability of the method to improve cloud analyses and GHI forecasts is demonstrated and a comparison using independent radiosoundings shows a reduction of specific humidity bias in the WRF analyses, especially in the low and mid troposphere. Ground-based GHI observations at 12 sites on Reunion Island are used to quantify the impact of CWP DA. Over a total of 44 days during austral summer time, when averaged over all sites, CWP data assimilation has a positive impact on GHI forecasts for all lead times between 5 and 14 hours. Root Mean Squared Error and Mean Absolute Error are reduced

15 by 4 % and 3 % respectively.



## 1 Introduction

The ongoing global transition from conventional to renewable energy is accompanied by an expected increase in installed photovoltaic (PV) capacity (Schmela et al., 2018). As an intermittent source of energy, PV requires solar irradiance forecasts in order to ensure grid stability and to enable an extensive feed-in of solar power into the electricity grids (Diagne et al., 2013).

The high solar potential in the tropics promises high yields of PV power. At the same time, it is particularly challenging to forecast global horizontal irradiance (GHI) in these regions. One example is the French overseas territory of Reunion Island which is located in the South-West Indian Ocean (SWIO) (Fig. 1). In this region, enhanced convection often causes large diurnal variability in solar irradiance (Badosa et al., 2013). Additionally, homogeneous, unstable air masses make it difficult to forecast convective initiation, cloud generation and cloud evolution. Consequently, solar irradiance forecast errors are especially

pronounced in the austral summer season (December-February) when convection is strong compared to winter (Badosa et al., 2015). Moreover, the specific topography of Reunion Island, with an elevation of up to 3069 m, results in an interplay of both breeze-induced clouds and orographic clouds due to the predominant south-easterly trade winds which are often extremely unpredictable.

    Numerical weather prediction (NWP) models are appropriate to forecast clouds and solar irradiance for lead times of more

than 6 hours ahead (Sengupta et al., 2017). Global circulation models (GCMs) currently provide weather information with an update frequency of four times per day, a temporal resolution of typically one hour and spatial resolutions of at least 10 km. In contrast, limited area models (LAMs) can provide weather parameters at higher spatio-temporal resolution and with increased frequency, which fits better to the requirements of the PV industry. Another benefit of using LAMs for solar power forecasting is that the choice of the parameterisation schemes of such a model can be optimised for a certain geographical region and the

model can be adapted to specific local PV forecasting requirements (López-Coto et al., 2013; Pérez et al., 2014).

    In general, NWP models tend to underestimate cloud cover, especially in the case of low clouds in coastal regions (Ruiz-Arias et al., 2016; Yang and Kleissl, 2016; Haiden and Trentmann, 2015), and therefore they often overestimate surface solar irradiance. The accuracy of cloud cover forecasts is limited by the predictability of clouds, the skill of the NWP model and its parameterisation schemes, and the quality of the initial conditions. In the case of LAMs, initial conditions can either be

derived directly by downscaling the GCM information or by applying data assimilation (DA) methods that statistically combine observations and background information such as previous forecasts (Kalnay, 2003).

    In regions where conventional observations (e.g. synop stations, ships, radiosoundings, etc.) are rare, geostationary meteorological satellites provide valuable information that can be used for DA within LAMs. The assimilation of satellite-derived cloud information into LAMs can be categorised into either radiance, or cloud property retrieval assimilation methods. Kurzrock et al.

(2018) provide a review of geostationary meteorological satellite DA in LAMs and show that evaluations of the diverse methods that focus on cloud analyses under tropical conditions are rare in peer-reviewed literature. Moreover, many methods make use of additional observations such as ground-based observations or radiosoundings in order to optimise the DA performance. This is not feasible in regions where conventional observations are sparse, such as the SWIO.



In the case of radiance DA it is challenging to assimilate cloud-affected observations due to issues of nonlinearity and uncertainty linked with moist processes. As a result, cloud-affected measurements are often excluded by such methods. Cloud property DA methods on the other hand explicitly focus on retrievals in regions of cloud presence in order to directly influence and improve the cloud analyses. Several considerations from Kurzrock et al. (2018) inform the methodology applied in this

study. Firstly, it is clear that the use of ensembles has become increasingly favoured, demonstrated by the growing number of hybrid DA methods. Secondly, the Weather Research and Forecasting (WRF) model (Skamarock et al., 2008) is the most widely applied LAM regarding both radiance and cloud property assimilation in peer-reviewed literature. Thirdly, it was shown that although multi-layer cloud property products from geostationary satellites do exist, they have been largely neglected so far by cloud property DA methods.

One example of such a product is the suite of cloud properties generated by NASA Langley's Satellite ClOud and Radiation Property retrieval System (SatCORPS, Minnis et al. (2016)). The SatCORPS is used to provide both gridded and pixel-level cloud retrievals in near real-time and post facto from multiple geostationary and polar-orbiting satellites. In this study, gridded SatCORPS Meteosat-8 cloud properties are assimilated for the first time.

For the aforementioned reasons, among the existing cloud property assimilation methods, the method of Jones et al. (2013)

has been identified as one of the most innovative and promising ones for short-term cloud cover forecasting. These authors develop a forward operator for cloud water path (CWP) retrievals for the Data Assimilation Research Testbed (DART) (Anderson et al., 2009). CWP is the column-integrated amount of cloud water in the form of liquid or ice that is bound between a cloud base pressure (CBP) and a cloud top pressure (CTP). The forward operator integrates the model mixing ratios of water, ice, graupel, rain and snow following the definition of Otkin (2010) to convert the column values of CWP into vertical distributions

of water. The application of this forward operator in DART was shown to improve cloud forecasts in two case studies of severe weather events over the United States (Jones et al., 2013, 2015).

Here we apply the CWP forward operator to SatCORPS gridded retrievals of three phases: liquid water path (LWP), supercooled water path (SWP) and ice water path (IWP). These retrievals are derived from the Spinning Enhanced Visible and InfraRed Imager (SEVIRI) sensor aboard Meteosat-8.

This work is the first to evaluate the impact of geostationary satellite DA on 1) LAM cloud representation in the SWIO and 2) short-term GHI forecasts for Reunion Island using an innovative cloud property DA method. This contributes to an assessment and quantification of the impact of satellite-based cloud observation DA with LAMs in the tropics.

Sect. 2 introduces the methods and data used and results are presented in Sect. 3 which consists of an evaluation of the DA cycling, a case study to investigate the link between DA and free forecasts of GHI, and an analysis of solar irradiance forecasts

for a total of 44 days. Conclusions are drawn in Sect. 4 along with an outlook for future work.





## 2   Methods and data

### 2.1   Model and cycling configuration

In this study the LAM used is the WRF model in its Advanced Research WRF (ARW) version 3.9.1.1. A single WRF domain is applied over a part of the South-West Indian Ocean including Reunion Island and Mauritius in its centre (Fig. 1). A horizontal
grid spacing of 12 km and 61 vertical levels stretching from the surface up to 50 hPa are used. We chose not to perform convection permitting simulations or nesting as our own investigations over Reunion Island, and studies of other regions (Lara-Fanego et al., 2012; Zhou et al., 2018) have shown that increasing the WRF grid spacing does not necessarily improve the performance of irradiance forecasts. A potential reason for this is that nonlinearity increases with increasing model grid spacing (Mass et al., 2002; Hohenegger and Schär, 2007). Moreover, DA with a two-way nested domain is considerably more
complex since the updated analyses of the nested domains must be physically consistent with those of the parent domains.

The Global Ensemble Forecast System (GEFS), which has a grid spacing of 0.5°, 27 vertical levels and a temporal resolution of three hours, consists of 21 ensemble members and provides WRF with boundary conditions (BCs) and initial conditions (ICs). In line with previous DA studies with LAMs, which use around 40 members (Schraff et al., 2016; Pan et al., 2014; Dillon et al., 2016), an ensemble consisting of 41 WRF members is applied in this study. The BCs and ICs of members 1-20 of the
WRF ensemble are generated from the original 20 GEFS members. In order to obtain 40 members using GEFS, the BCs and ICs of members 1-20 are perturbed with the WRF Data Assimilation (WRFDA, Barker et al. (2012)) system using the standard NCEP background error covariance to generate WRF members 21-40. The Global Forecast System (GFS) with a grid spacing of 0.25°, 32 vertical levels and a temporal resolution of one hour is used for the BCs and ICs of the 41st WRF member. Initial conditions for all members are only generated from GEFS and GFS at the start of the DA cycling after which they are fed in
from the previous cycling step.

All members use the same model configuration, this includes the Thompson microphysics scheme (Thompson et al., 2008), the Dudhia scheme for shortwave radiation (Dudhia, 1989), and the Kain-Fritsch scheme (Kain, 2004) for cumulus parameterisation. These schemes are among the most commonly used in WRF configurations according to a WRF physics survey (UCAR, 2015) and this configuration also performed well for Reunion Island in preceding experiments (not shown).

Due to the higher spatio-temporal resolution of the BCs of the GFS member, after each DA step the first 40 members of the ensemble are re-centred on the GFS member (member 41). The ensemble mean is subtracted from each member to obtain the perturbations from the mean for each member. These perturbations are then individually added to the 41st member to re-generate the 40 member ensemble, which now has member 41 as its mean.

The DA cycling interval is 6 hours leading to 4 cycle steps per day, i.e. DA is performed at 0000, 0600, 1200 and 1800 UTC
leading to new analyses (also referred to as "posteriors") at these times from which first guess forecasts (also referred to as "priors") for a lead time of 6 hours are performed. As satellite observations are available more frequently than every 6 hours it is possible to reduce this interval to provide updated analyses for free forecasts (FF) at a higher update rate than 6 hours. For reasons of simplicity, this study uses 6-hourly cycling.





Since member 41 uses the BCs with the highest spatio-temporal resolution this member is chosen for free forecasts that are realised every 24 hours at 0000 UTC (0400 local time) using its updated analyses (ICs) from the DA cycling. The BCs for the free forecasts are likewise provided by GFS and are generated for lead times up to 16 hours.

Two cycling and free forecast experiments were performed: A cycling experiment with CWP DA (CWPDA) and a control
cycling experiment without DA where all observations are only evaluated by DART but not assimilated (CTRL). The respective free forecast experiments that are run using the 0000 UTC analyses of the two cycling experiments are labelled CWPDA-FF and CTRL-FF.

The experiments of this study are performed for the austral summer of 2017 and 2018, i.e. between 9 December 2017 and 1 March 2018. Some periods are excluded in the simulations due to gaps in CWP data availability and cyclonic activity.
Therefore, cycling is performed for several smaller periods, listed in Table 1). The convective activity is generally pronounced during these periods and in each case at least two DA steps were performed before using a 0000 UTC analysis for the free forecasts periods listed in Table 1.

## 2.2 CWP assimilation methodology

The DA cycling is performed using the DART and its Ensemble Adjustment Kalman Filter (Anderson, 2001) and the CWP
forward operator developed by Jones et al. (2013) is used to assimilate CWP observations. Jones et al. (2015) describe a few updates to the forward operator, which is close to the version used in this study.

In comparison to other cloud property assimilation methods that do not take into account multi-level cloud information, one strength of this forward operator is that it accounts for cases when the model and CWP observations contain clouds that are localised at different altitudes. The forward operator does this through adjustments to the modelled CWP. For example, if the
model contains a low level cloud and the observations indicate high level cirrus, the integrated CWP value might be similar and the impact to the model analysis would be small if the cloud altitude was not considered. In this case the forward operator constrains the model CWP to the level of the observed cloud, leading to a larger impact in the analysis and a cloud is introduced at the correct vertical location.

The SatCORPS cloud products are retrieved using algorithms originally developed to analyse MODerate-resolution Imaging
Spectroradiometer (MODIS) aboard Terra and Aqua for the NASA Clouds and the Earth's Radiant Energy System (CERES) project (Minnis et al., 2011; Trepte et al., 2018). These algorithms have been adapted to other imagers aboard geostationary (Minnis et al., 2008) and other low Earth-orbit satellites (Minnis et al., 2016). Among other parameters, the SatCORPS data set include both a pixel and a gridded CWP product.

Depending on the cloud top phase, as determined from Cloud Top Temperature (CTT), a given pixel is defined as either ice,
supercooled liquid or liquid under the assumption that the cloud phase and particle size are vertically homogenous. The actual CWP is derived as a function of total optical depth and particle size retrievals. At night, the retrievals of cloud optical depth are not as accurate as during daytime due to the lack of visible data.

The global gridded data have a grid spacing of 0.25° (approximately 28 km) latitudinally and 0.3125° (approximately 34 km) longitudinally and are independent of the respective satellite sensor resolution (i.e. the resolution of the pixel product). The




data are available at hourly resolution. For this study, the retrievals of Meteosat-8 are used. The satellite has been operational over the Indian Ocean since February 2017 and has a sub-satellite position of 41.5° East.

SatCORPS provides water path (WP) retrievals in three phases: liquid water path (LWP), supercooled water path (SWP) and ice water path (IWP). The supercooled phase is determined using a post retrieval classification. If the pixel phase is liquid and has a CTT below 273.15 K the pixel phase is defined as supercooled liquid. The retrievals of the different phases are assimilated as independent observations using the same forward operator which can be considered as multi-layer cloud property assimilation.

Each of the three-phased WP retrievals are bound between a cloud base pressure (CBP) and a cloud top pressure (CTP) that are also included in the SatCORPS product. The forward operator requires information about the cloud effective pressure (CEP) for vertical localisation. Following (Jones et al., 2015) we set this to be the mean of CTP and CBP.

As this is the first time that SatCORPS gridded Meteosat-8 retrievals are assimilated, a horizontal localisation radius has to be defined for these observations. The Gaspari-Cohn covariance cutoff method is used for horizontal localisation (Gaspari and Cohn, 1999). In this study, the half-width of the localization radius (also called cutoff) is set to 90 km, a value that performed well in sensitivity runs (not shown). Consequently, there is a factor 3 difference between observation grid spacing (approximately 30 km) and cutoff. This factor corresponds approximately to the factor 3.3 applied by Jones et al. (2015) who use a cutoff of 20 km for satellite observations at 4 km nominal resolution.

While in radiance assimilation the historical approach is to assimilate only clear sky observations, avoid cloud-affected observations and gradually move towards all-sky assimilation (Kurzrock et al., 2018), an opposite strategy is followed in this work on cloud property assimilation. No clear sky retrievals are assimilated in the experiments in this study. Test experiments showed that the assimilation of clear sky retrievals led to a strong reduction in the amount of cloud in WRF simulations with the current experimental setup. This means that defining a minimum threshold for 'cloudy' WP retrievals allows the EnKF to predominantly assimilate observations over cloudy locations. To address cases when WRF tends to underestimate cloud presence we use a minimum threshold of $0.04 \, \mathrm{kg \, m^{-2}}$ to define cloudy observations of all phases.

We apply the same retrieval errors as Jones et al. (2013, 2015) (Table 2). These were defined for the Geostationary Operational Environmental Satellite (GOES) derived pixel data of the SatCORPS product over the US for both IWP and LWP. There is yet to be a study assessing the errors of the gridded product for Meteosat-8 based retrievals. Therefore, these errors serve as a first estimate for our study. The true uncertainty in CWP varies with cloud conditions, solar and viewing geometry and other factors which need to be assessed more thoroughly. Defining more region specific errors as well as independent errors for each phase may be an objective of future work.

Spatially-varying state space adaptive covariance inflation is applied to the first guess at each cycling step (Anderson, 2009, 2007). This is a commonly used method to increase prior ensemble spread and prevent the ensemble from collapsing to a single solution.



### 2.3 DA evaluation using radiosoundings

Independent observations are used to evaluate the DA method in addition to the assimilated observations. The WRF domain in this study contains one radiosonde station at Gillot-Aeroport (Fig. 2) where radiosondes are launched daily at 1200 UTC. These soundings provide valuable independent in-situ observations that are used for observation space diagnostics. The soundings

are available via the National Center for Environmental Prediction (NCEP) Global Data Assimilation System (GDAS) data set and are evaluated within DART.

### 2.4 Solar irradiance forecast verification

An evaluation of the GHI forecasts produced is performed using pyranometer observations provided by Météo-France from 12 locations spread across Reunion Island (Fig. 2) at various altitudes between 5 m (Pointe des Trois-Bassins) and 2149 m (Piton-

Maido). The raw observations have a temporal resolution of 6 minutes and a linear interpolation to 15 minutes is performed to match the WRF output.

The quality control measures for GHI observations consist of a visual verification and application of the sub-hourly data quality control procedures proposed by Espinar et al. (2012) that detect extrema, rare observations and maximum steps for two following measures.

It is common to perform spatial averaging of WRF solar irradiance output around the site of interest rather than using only the GHI forecasted for the closest grid box (Verbois et al., 2018; Lara-Fanego et al., 2012; Zhou et al., 2018). This reduces the variability of the forecasted GHI and thus typically reduces forecast errors in terms of the standard metrics, Root Mean Squared Error (RMSE) and Mean Absolute Error (MAE). For each GHI forecast at a given location and lead time we apply inverse distance weighting (IDW) (Shepard, 1968) according to the formula:

$$\widehat{GHI(b_c)} = \frac{\sum_{i=1}^{n} GHI(b_i) d_i^{-1}}{\sum_{i=1}^{n} d_i^{-1}} \tag{1}$$

where $d_i$ is the distance between the grid box containing the observation site, $b_c$, and another grid box, $b_i$. The best results were seen when $b_i$ was considered within a 75 km radius of the observation site. Besides IDW, no further post-processing is applied to the WRF solar irradiance forecasts.

Aerosol optical thickness in the study region rarely exceeds 0.2 (Stöckli, 2018) leading to a stable clear sky irradiance on

Reunion Island as compared to other regions of the world where aerosol optical thickness is highly variable. Moreover, the influence of volcanic aerosols on solar irradiance can be neglected as there were no volcanic eruptions on Reunion Island during the overall study period. We therefore assume that fluctuations between forecasted and observed GHI caused by aerosols are negligible compared to the influence of clouds.

The clear sky model of the European Solar Radiation Atlas (ESRA) (Rigollier et al., 2000) is used for clear-sky normalisation

of solar irradiance forecasts.





## 2.5 Validation metrics

Various metrics are considered to evaluate the performance of WRF outputs as compared to CWP retrievals, radiosoundings and GHI observations. For $n$ predictions, $y$, of the observation ($y$ can be either a prior or posterior) and observations, $o$, these metrics are defined as follows. The RMSE is defined as:

$$RMSE = \sqrt{\frac{1}{n}\sum_{i=1}^{n}(y_i - o_i)^2} \qquad (2)$$

The MAE is defined as:

$$MAE = \frac{1}{n}\sum_{i=1}^{n}|y_i - o_i| \qquad (3)$$

MAE becomes the Mean Bias Error (MBE or simply bias) if the absolute of $y_i - o_i$ is not taken in equation 3.

The correlation (or Pearson correlation coefficient) is defined as:

$$corr_{y,o} = \frac{cov(y,o)}{\sigma_y \sigma_o} \qquad (4)$$

with $cov$ being the covariance and $\sigma$ being the standard deviation.

The total spread (TSPRD) is defined as the pooled spread of the ensemble and observation errors:

$$TSPRD = \sqrt{\frac{\sigma_y + \sigma_o}{2}} \qquad (5)$$

with $\sigma_o$ being the standard deviation of the observation error and $\sigma_y$ being the spread of the 41-member ensemble which is defined as:

$$\sigma_Y = \sqrt{\frac{1}{41}\sum_{m=1}^{41}(Y_m - \overline{Y})^2} \qquad (6)$$

where $\overline{Y}$ is the ensemble mean.

## 3 Results

### 3.1 Cycling evaluation

This subsection evaluates the implementation of the CWP DA methodology.





As a first step, the RMSE for WP at different altitudes during all cycling periods listed in Table 1 is calculated and shown in Fig. 3 in which the impact of the three-phased CWP assimilation can be seen. For each phase and altitude, the majority of the available observations (green circles) are assimilated (green asterisks), which indicates that the DART quality check only excludes a reasonable amount of observations and that the defined WP errors (Table 2) are realistic. Depending on the mean

CEP of the different phases, the maximum number of WP observations, approximately 20000 per phase, are localised around certain pressure level bins. These bins are 400 hPa for IWP, 500 hPa for SWP and 850 hPa for LWP.

The difference between the first guesses (solid lines) and the analyses (dashed lines) is a measure of the impact of the respective phase on the analysis and it can be seen that IWP has the greatest impact and LWP has the lowest impact. This is to be expected since IWP usually exhibits the largest absolute values of WP. This difference in absolute values also affects the

RMSE which, at 400 hPa, is highest for IWP with averages of 0.43 $\mathrm{kg\,m^{-2}}$ and 0.13 $\mathrm{kg\,m^{-2}}$ in the first guesses and analyses respectively. In comparison, the RMSE for LWP at 850 hPa is reduced from 0.09 $\mathrm{kg\,m^{-2}}$ in the first guesses to 0.08 $\mathrm{kg\,m^{-2}}$ in the analyses.

The evolution of RMSE, MBE and TSPRD for the ensemble mean first guess and analysis is shown in Fig. 4. Cycling period C (Table 1) is chosen as an example and the pressure levels identified above are chosen for the respective phase.

The classical sawtooth pattern caused by the differences between first guess and analysis is most clearly visible for IWP. This indicates that the retrievals of this phase have the most impact in the filter. While a reduction of RMSE between first guess and analysis is visible at most assimilation times for SWP, this is not the case for LWP. In fact, the smallest impact is achieved for LWP, which is in line with the findings shown in Fig. 3.

TSPRD and RMSE are roughly in phase for IWP, the values generally increase to around 0.3 $\mathrm{kg\,m^{-2}}$ for IWP in the 6 hour

first guess forecasts. Relatively high RMSE values of more than 0.6 $\mathrm{kg\,m^{-2}}$ are reached twice for IWP during this period, but these are linked to exceptionally large biases in both cases. In the analyses RMSE and TSPRD are mostly around 0.1 $\mathrm{kg\,m^{-2}}$.

RMSE and TSPRD are less in phase for SWP, which is in line with the expected lower impact of this phase. For LWP, TSPRD never assumes values lower than 0.05 $\mathrm{kg\,m^{-2}}$ which is the defined observation error for the lowest WP observations (Table 2). This underlines once more the importance of determining phase-dependent errors that should ideally be less than

0.05 $\mathrm{kg\,m^{-2}}$ for low LWP retrievals. As can be seen in Fig. 4, a clear difference between first guess and analysis becomes visible only at times where TSPRD is larger than 0.05 $\mathrm{kg\,m^{-2}}$. Thus, the relatively large observation error for the smallest observations is likely the reason for the comparably small impact of LWP.

The number of assimilated observations may be very different than the number of available observations due to quality control within DART, as can be seen in Fig. 3. The example of SWP in Fig. 4 shows that the number of assimilated observations

is not necessarily correlated with an improvement in terms of RMSE. For example, on the 12th of January the number of assimilated observations fluctuates between 80 and 300 but the RMSE remains relatively stable. For all three phases, the number of assimilated observations per assimilation time varies heavily from zero to a few hundred observations depending on the time of day.

The MBE for IWP fluctuates around zero with a positive correction in each analysis. For SWP and LWP there is a contin-

uously negative MBE, mostly between -0.1 $\mathrm{kg\,m^{-2}}$ and -0.05 $\mathrm{kg\,m^{-2}}$. This indicates that WRF underestimates clouds in the




middle and lower troposphere. DA corrects for MBE for all three phases but is far from achieving MBEs close to zero for SWP and LWP.

Figures 3 and 4 show evaluations of DA experiments as compared to the assimilated observations and therefore demonstrate the ability of the DA cycling to constrain the WRF simulations. Although this can be used to assess the performance of the assimilation itself, this cannot act as an exhaustive validation as no independent observations are used.

Consequently, an evaluation of the MBE for temperature and specific humidity using independent radiosonde measurements is shown in Fig. 5. All 43 radiosoundings at Gillot-Aeroport during the complete study period (Table 1) are considered. The total number of evaluated observations per pressure bin are shown in green and reach up to 80 at around 400 hPa. The lack of radiosonde stations in the model domain and the fact that only one station in the centre of the domain is considered is compensated, to some extent, by the duration of the study period that includes various cloud and weather situations.

The ensemble mean first guess (solid lines) and analysis (dashed line) are shown for both experiments CTRL (orange line) and CWPDA (black lines) with first guess and analysis being the same for CTRL as no DA is performed. For both experiments an overall negative MBE for specific humidity and a positive MBE for temperature are visible throughout the troposphere. The fact that the MBE for specific humidity is largest in the lower troposphere, with more than $1\,\mathrm{g\,kg^{-1}}$, confirms that WRF tends to underestimate low clouds. At the same time, the difference between the two experiments for specific humidity is the largest in the lower troposphere around 850 hPa. This shows that the assimilation corrects for the lack of humidity in the analyses to some extent and thus has the effect of a bias correction.

Although Fig. 3 and Fig. 4 indicate the largest impact of the DA is in the higher troposphere from IWP observations, the effect on bias regarding radiosonde specific humidity is smaller at these altitudes than in the low troposphere. This may be explained by the fact that absolute values of specific humidity are generally largest in the low troposphere. Moreover, the evaluated radio soundings are valid only for the centre of the domain while Fig. 3 and Fig. 4 include information about the whole model domain. Furthermore, local thermal circulations likely cause more low clouds at this coastal location than in the rest of the domain which lacks other land masses.

The difference between CWPDA prior and posterior is more distinct for specific humidity than for temperature leading to an improvement of humidity bias in the analyses compared to the first guesses. As the objective here is to improve cloud prediction, the improvement in humidity, a field strongly related to cloud, is more significant than an improvement in temperature. It is however favourable that the CWP DA does not have a negative impact on the temperature profile.

Regarding the DA configuration, a number of parameters such as the covariance localisation radius are known to largely impact the DA outcome (Otkin, 2012; Ying et al., 2018). Concerning the gridded multi-phase SatCORPS retrievals used in this study, the sensitivity of the assimilation to the localisation radius might be assessed in detail in future work. This is especially true for experiments at convective scale resolutions that have yet to be performed.

Moreover, the ensemble spread could be modulated in various ways by adjusting the WRF ensemble generation method. For example, multiple sets of physics options could be applied in the ensemble members, an optimal compromise between ensemble size and spread could be determined, the method for WRF ensemble member generation from the 21-member GEFS





ensemble ICs and BCs might be optimised, and different settings for adaptive inflation could be tested. The WRF ensemble may also be applied to the free forecasts for probabilistic solar irradiance forecasting.

In summary, the operational correctness of the DA methodology is confirmed. The largest impact is found for the ice phase retrievals, and the lowest impact for the liquid phase retrievals. Independent radiosoundings indicate a humidity bias reduction

between first guesses and analyses. Any improvement of the utilised cloud products is expected to positively influence the DA outcome. Future cloud products are aiming to better account for vertical heterogeneity and thus produce multi-phase CWP estimates that are closer to reality and more similar to what NWP models produce regarding deep overlapping cloud systems. Moreover, a precise definition of phase-dependent errors for the SatCORPS Meteosat-8 products does not yet exist. Once this information is obtained, the performance of the applied system using these observation errors can be assessed.

## 3.2 Case study evaluation

Having demonstrated the correct implementation of the DA methodology, which shows a generally positive impact on the cloud analyses, this subsection focusses on a case study of one particular day. The influence of the applied DA strategy on the cloud analysis and the subsequent free forecast with respect to solar irradiance is analysed.

On 12 January 2018 the large scale flow in the study region was governed by an anticyclone south of Madagascar and

two depressions located at the northern boundary of the WRF domain as indicated by synoptic surface analyses published by Météo-France and the South African Weather Service (not shown). The first depression was located approximately 700 km north of Reunion Island, the second approximately 1500 km north-east of Reunion Island. This situation led to a north-westerly flow throughout the model domain and the creation of a convergence zone extending diagonally across the model domain from the north-west to the south-east.

Such large convective cloud systems associated with low pressure systems north of Reunion Island typically produce the lowest GHI values during austral summer (Badosa et al., 2015). Hence, the day considered here has one of the lowest observed GHI values throughout the combined study period listed in Table 1 and therefore most distinctly shows the impact of the DA on the GHI forecast.

The left part of Fig. 6 shows SatCORPS WP retrievals at the time of the analysis (0000 UTC) of the free forecasts performed

for this day. The clouds induced by the convergence zone are visible especially in the ice and liquid phase, with WP values exceeding $0.3 \ \mathrm{kg \, m^{-2}}$ in some areas. The highest values in the vicinity of Reunion Island can be observed in the ice phase indicating high clouds. These ice clouds persist during the day (not shown) and contribute largely to the observed low solar irradiance on Reunion Island throughout the day.

The right part of Fig. 6 shows observation space diagnostics of the WP difference, in each phase, between the two experi-

ments (CWPDA minus CTRL) in terms of the posterior ensemble mean WP. More cloud water is present in all phases over the convergence zone in CWPDA when compared with the CTRL experiment. In the ice phase, distinct gradients between areas of increased and decreased WP are visible that indicate corrections to the cloud location resulting from the DA. Maximum and minimum values are -0.9 and $1.9 \ \mathrm{kg \, m^{-2}}$ for IWP (b), -0.2 and $0.19 \ \mathrm{kg \, m^{-2}}$ for SWP (d) and -0.06 and $0.19 \ \mathrm{kg \, m^{-2}}$ for LWP (f).



The effect of these corrections on the free forecast experiment in terms of cloud fraction is shown in Fig. 7. The ice clouds induced by the convergence zone are visible in both CTRL-FF and CWPDA-FF and in both cases, clouds are located above Reunion Island 15 minutes after the analysis time ((a) and (b)). A north-eastward relocation of the clouds around Reunion Island is visible for CWPDA-FF. As the clouds move eastward over the course of the day, this has the effect of causing high clouds

to persist over Reunion Island in CWPDA-FF ((d) and (f)) while clouds are further west in CTRL-FF ((c) and (e)) leaving the model levels around 300 hPa cloud-free above Reunion at 0800 UTC (corresponding to noon local time). In CWPDA-FF, clouds are still present at 300 hPa over Reunion Island at 0800 UTC leading to more realistic forecasts of solar irradiance when compared with ground observations as shown in the following.

A large variability of GHI is observed during the day for all pyranometer sites as well as between the different sites (Fig. 8).

GHI values are overall low and mostly below $400 \, \mathrm{W \, m^{-2}}$ at all sites, this is mainly caused by the high clouds during that day as determined from satellite images. A distinct difference is visible between CTRL-FF and CWPDA-FF as a consequence of the DA. Although the forecasted GHI is largely reduced in CWPDA-FF, the values around noon are still too high compared to the observations for most sites. Further improvements might be found by analysing the interplay between DA, post-processing and the configuration of WRF in terms of grid spacing, nesting and the choice of parameterisation schemes. In this study we

focus on the influence of DA only which is clearly visible in this example.

This comparison between forecasts and observations of GHI also illustrates the difficulty of forecasting ramp events. The chosen grid spacing of 12 km and IDW post-processing results in a smoothing of ramps in the WRF forecasts, which is why the widely used metrics RMSE and MAE are suitable for a quantification of the DA impact. If one wants to study the impact of DA on ramp forecasts specifically, experiments at convection resolving resolutions, a focus on parameterisation schemes,

and specific ramp metrics and post-processing methods are required.

### 3.3 Free forecast evaluation

The previous subsection illustrates that cloud-property DA can have a considerable positive impact on short-term forecasts of cloud-related parameters such as CWP, specific humidity and solar irradiance in the SWIO. Nevertheless, as in many other proofs of concept for DA methods, only one weather situation has been considered so far in terms of solar irradiance. An

evaluation over a period of more than a few days is rare in peer-reviewed literature (Kurzrock et al., 2018). Therefore, an evaluation of the free forecasts for a total of 44 days (Table 1) is performed in this subsection in order to quantify the impact of DA on GHI forecasts more meaningfully.

Figure 9 shows an evaluation of GHI forecasts from CTRL-FF and CWPDA-FF for the 12 considered sites at Reunion Island. The spread between the sites for all metrics can be explained by the location of the sites and the local meteorological

conditions. The six sites with the best performance in terms of RMSE (between 230 and 280 $\mathrm{W \, m^{-2}}$) and MAE (between 180 and 220 $\mathrm{W \, m^{-2}}$) are Gillot-Aeroport, Pierrefonds-Aeroport, Pointe des Trois-Bassins, Piton Sainte-Rose, Le Port and Le Baril. All of these sites are located on the coast line of Reunion (Fig. 2) where the influence of clouds that are induced by orographic uplift and thermal circulations caused by the mountains is lowest.



The two sites with the worst results in terms of RMSE, MAE and correlation are Colimacons and Petite-France. These sites are both located in the west of the island at 800 m and 1200 m respectively, meaning these sites are typically in the lee of the trade winds at altitudes where thermally driven convective clouds often form. This leads to lower GHI values (Badosa et al., 2013) and produces the most complex solar irradiance conditions (Bessafi et al., 2018) compared to the other sites. This

explains the high positive MBE for these two sites.

A positive bias can be seen for most sites, which confirms that WRF tends to overestimate GHI on Reunion Island during summer time with pronounced convective activity. There is a shift to lower values of MBE between CTRL-FF and CWPDA-FF with approximately the same amplitude for all sites. This illustrates that CWPDA-FF generally includes more clouds than CTRL-FF but it also means that for sites with a negative MBE, typically the ones that are not located in the mountains, there is

a degradation towards more negative values of MBE.

An improvement of GHI forecasts between CTRL-FF and CWPDA-FF is visible for almost all sites in terms of RMSE, MAE and correlation. On average across all sites, RMSE improves by 11 $\mathrm{W\,m^{-2}}$ (4 %), MAE by 6 $\mathrm{W\,m^{-2}}$ (3 %) and correlation by 0.03. The only exception to this is the site at Le Port where the RMSE is degraded by 1 $\mathrm{W\,m^{-2}}$ and MAE by 2 $\mathrm{W\,m^{-2}}$. In terms of RMSE and MAE, there is less improvement at sites for which GHI is predicted most accurately (Gillot-Aeroport and

Pierrefonds-Aeroport). This may be explained in the same way as the large improvement at the sites with the least accuracy: As found above, DA leads to both a better representation of cloud location, and generally increased lower tropospheric moisture. Both of these effects have an impact on the sites with the least accuracy. Being located far from mountain slopes, thermally induced convective clouds are rarer at Gillot-Aeroport and Pierrefonds-Aeroport. The effect of low level clouds at the mountain slopes is thus of less consequence here, compared to improved large-scale cloud system locations.

Figure 9 shows the averaged intraday values for each evaluation metric over all forecasted lead times. The RMSE per lead time as a mean over all sites is shown in Fig. 10 in order to evaluate the impact of the applied DA method on the free forecast of GHI with respect to lead time.

GHI observations are not always available for all 12 sites for a given lead time and day which complicates comparisons of different lead times. Consequently, the number of considered days per lead time shown in the figure is the number of days for

which observations are available for all 12 sites. These are the days that have been considered in the calculation of RMSE for a given lead time. The first lead time is no earlier than 5 hours (0500 UTC or 9 am local time) since the observations in the morning did not pass the quality control for several stations, which is often due to shadowing, mainly caused by the mountains.

The free forecasts that have been initialised with CWPDA have a lower RMSE than CTRL-FF throughout the day. Both the absolute (a) and the clear-sky normalised RMSE (b) are shown in Fig. 10. The representation of absolute RMSE does

not correct for the diurnal cycle of GHI which leads to the characteristic curve. It does however allow the identification of the absolute difference in RMSE between CTRL-FF and CWPDA-FF that reaches up to 60 $\mathrm{W\,m^{-2}}$ at 0800 UTC (noon local time).

The normalisation with the clear sky irradiance removes the diurnal cycle showing an expected increasing forecast error with increasing lead time for both experiments. The maximum absolute difference of 60 $\mathrm{W\,m^{-2}}$ between CTRL-FF and CWPDA-

FF translates to approximately 2 % of normalised RMSE (Fig. 10 (b)).





With increasing lead time, the predictability of cloud evolution decreases and the influence of the boundary conditions is expected to become larger compared to that of the initial conditions. This means that the differences between CTRL-FF and CWPDA-FF are expected to be larger for short lead times than for longer ones, indicating most DA impact occurs close to the analysis time. According to Fig. 10 this is not the case in the present experiments, rather, in the first 14 hours of free forecast

the difference between CTRL-FF and CWPDA-FF does not change remarkably except for the first 2 lead times (0445 UTC and 0500 UTC), for which RMSEs is almost identical.

Together with the above findings this indicates that the DA of CWP generally adds more clouds to WRF, reduces the overestimation of GHI, and therefore acts like a sophisticated bias correction. On average, the additional clouds do not vanish within the first 14 hours of forecast, leading to improved GHI forecasts throughout the day.

**4  Conclusions**

Previous studies have shown that the assimilation of geostationary CWP retrievals with WRF-DART leads to improved short-term GHI forecasts. However, this improvement has never been quantified for study periods of more than a few days. Moreover, the performance under tropical conditions has been unknown so far. Herein, the successful assimilation of multi-phase geostationary CWP retrievals with a 41-member WRF ensemble over the SWIO is demonstrated for a total of 44 days in autral

summer and the impact on short-term GHI forecasts for Reunion Island is quantified.

Gridded Meteosat-8 retrievals of liquid, supercooled liquid and ice water path from NASA Langley's SatCORPS are assimilated in a 6-hourly cycling procedure. Free forecasts using initial conditions from the cycling are produced once per day starting at 0000 UTC. Control experiments without DA are performed for both the cycling and free forecasts, enabling an evaluation of the impact of the applied DA methodology.

It is demonstrated that the assimilated retrievals of IWP, SWP and LWP have most impact at pressure levels of 400 hPa, 500 hPa and 850 hPa respectively. The largest contribution comes from the IWP retrievals with an average reduction in RMSE of approximately 0.2 kg m$^{-2}$ between first guess and analysis. LWP has the lowest impact which can partly be explained by the large observation errors for small observations.

The evaluation using 43 independent radiosoundings shows a reduced bias in specific humidity for the experiment with CWP

DA, especially in the mid-troposphere. A further reduction of bias between the first guesses and analyses supports the case that the applied DA method leads to more realistic WRF humidity profiles and consequently improves the 'cloud analyses'.

Although this DA methodology is yet to be fully optimised, the positive impact on short-term solar irradiance forecasts demonstrated in this paper is clear. The comparison between inverse-distance weighted WRF forecasts and ground-based observations of GHI at 12 sites on Reunion Island allows a quantification of the DA impact. The study period in austral

summer 2017/2018 includes complex irradiance conditions and enhanced convection compared to winter time. Two major effects of the applied method can be deduced. Firstly, the location of large-scale cloud systems is corrected, and secondly, the increased amount of lower tropospheric water in WRF leads to more breeze-induced convection on the slopes of the island.




The evaluation of GHI forecasts from the experiments without and with DA shows an overall reduction of 4 % for RMSE and 3 % for MAE due to CWP DA. The method of refining the ICs using CWP DA causes a positive impact on GHI forecasts for the whole duration of the forecast, i.e. up to a lead time of 14 hours.

There is a potential for improvements of the applied methodology, mainly regarding the DA configuration and CWP retrieval errors. As more geostationary satellites of the third generation become operational, the resolution of such observations increases and global gridded cloud products may be of higher resolution in the future. This would open new possibilities for multi-layer cloud information to be assimilated in a similar manner as presented here.

This work is a contribution to improved short-term solar irradiance forecasts in complex Tropical environments. The obtained results allow to produce more accurate solar power forecasts, and may have positive impacts on other applications that depend on accurate information about cloudiness. As the cloud products used here are available globally, the method offers a portable and globally applicable approach.

*Code and data availability.* The WRF-ARW source code is publicly available at http://www2.mmm.ucar.edu/wrf/users/. Version 3.9.1.1 of WRF-ARW (released on 28 August 2017) is used for this work. The DART source code (Anderson et al., 2009) including the CWP forward operator (Jones et al., 2013) is publicly available at https://www.image.ucar.edu/DAReS/DART/. The manhattan version of DART (release 12002) is used for this work. The SatCORPS Meteosat-8 retrievals (Minnis et al., 2016) are available in near real-time at https://cloudsway2.larc.nasa.gov/. GFS and GEFS data are available at http://www.ftp.ncep.noaa.gov/data/nccf/com/. The radiosoundings are contained in GDAS data available at https://nomads.ncdc.noaa.gov/data/gdas/. The solar irradiance observations used in this study are available via Météo-France (http://www.meteofrance.com) against payment.

*Author contributions.* HN collected GFS, GEFS and SatCORPS data, and set up the cycling DA experiment environment. JS performed data validation and visualisation. CL created the topographical maps and processed the GHI observations. FCM, SC, LL and GL supervised the research activity. FK devised the methodology, realised and evaluated the experiments, and wrote the paper. All authors were involved in discussions throughout the development and experiment phase, and all authors commented on the paper.

*Competing interests.* The authors declare that they have no conflict of interest.

*Acknowledgements.* We are grateful for the help of Nancy Collins and Glen Romine from the DART team at the National Center for Atmospheric Research. Moreover, we would like to thank Météo-France for providing us with the pyranometer observations via the Laboratoire de l'Atmosphère et des Cyclones.



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



**Table 1.** Overview of the cycling experiment periods and the associated free forecast experiment and dates.

| Cycling period | Cycling start (UTC) | Cycling end (UTC) | Dates of free forecast | number of days forecasted |
|---|---|---|---|---|
| A | 2017-12-09 12:00 | 2017-12-12 00:00 | 2017-12-10 – 2017-12-12 | 3 |
| B | 2017-12-19 00:00 | 2017-12-27 18:00 | 2017-12-20 – 2017-12-27 | 8 |
| C | 2018-01-08 12:00 | 2018-01-13 00:00 | 2018-01-09 – 2018-01-13 | 5 |
| D | 2018-01-20 12:00 | 2018-01-31 18:00 | 2018-01-21 – 2018-01-31 | 11 |
| E | 2018-02-11 12:00 | 2018-02-13 00:00 | 2018-02-12 – 2018-02-13 | 2 |
| F | 2018-02-14 00:00 | 2018-03-01 00:00 | 2018-02-15 – 2018-03-01 | 15 |



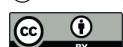

**Table 2.** Overview of the cycling experiment periods and the associated free forecast experiment and dates.

| WP thresholds ($\mathrm{kg\,m^{-2}}$) | Assigned WP errors ($\mathrm{kg\,m^{-2}}$) |
|---|---|
| 0.040 < WP < 0.050 | 0.050 |
| 0.200 < WP < 0.075 | 0.075 |
| 0.500 < WP < 0.100 | 0.100 |
| 1.000 < WP < 0.125 | 0.125 |
| 2.500 < WP < 0.150 | 0.150 |



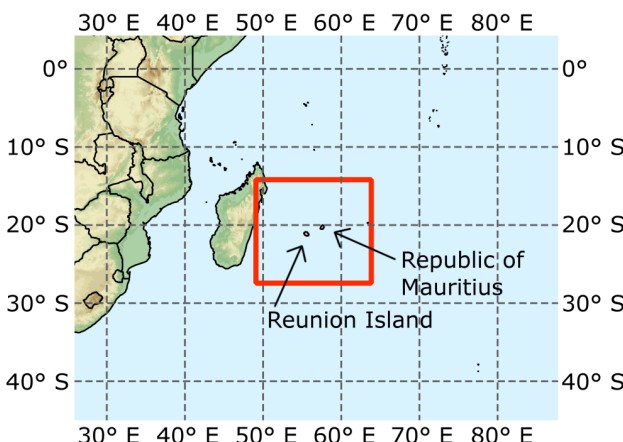

**Figure 1.** Study region and WRF domain (red rectangle) in the South-West Indian Ocean.



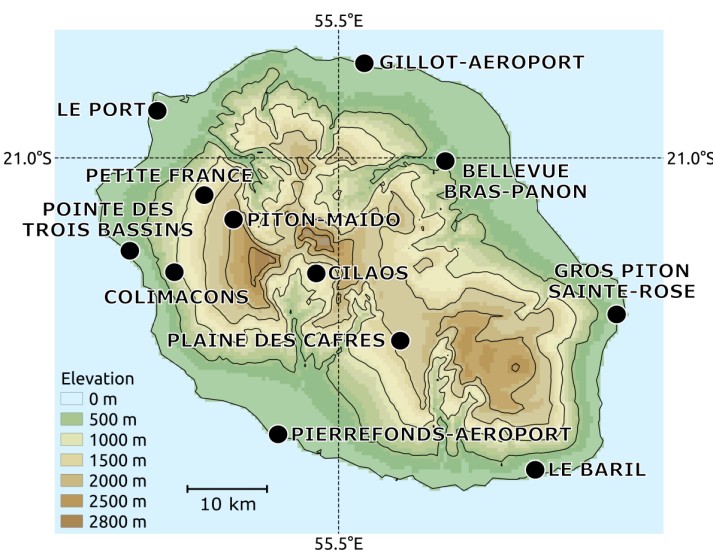

**Figure 2.** Topography of Reunion Island (21° S, 55.5° E) and the locations of the 12 Météo-France pyranometers used for the evaluation of the WRF solar irradiance forecasts.



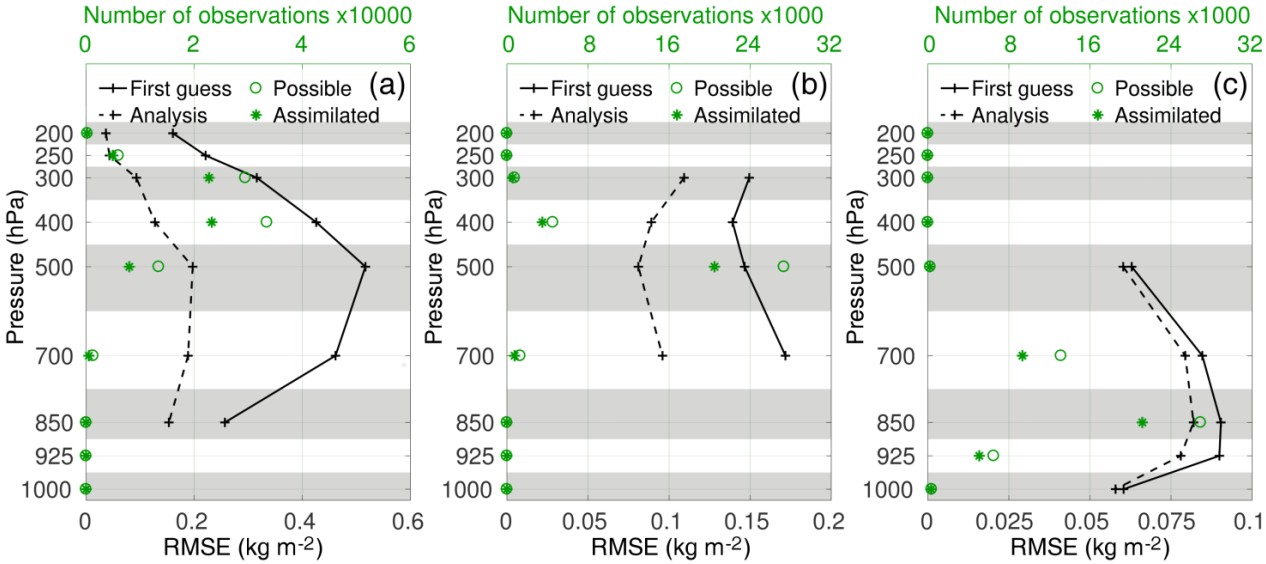

**Figure 3.** Vertical profiles of the RMSE per bins of CEP for the three phases IWP **(a)**, SWP **(b)** and LWP **(c)** for ensemble mean first guess (solid lines) and analysis (dashed lines) as a mean over all periods listed in Table 1. The number of possible (circles) and assimilated (asterisks) observations is shown in green.



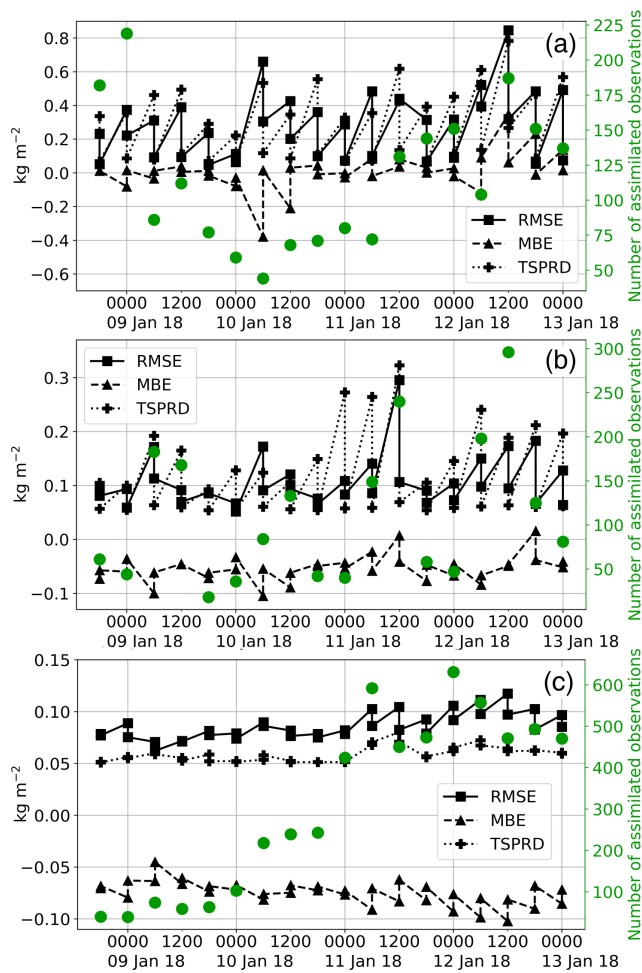

**Figure 4.** Temporal evolution of ensemble mean first guess and analysis for the three phases IWP at 400 hPa **(a)**, SWP at 500 hPa **(b)** and LWP at 850 hPa **(c)** for cycling period C (Table 1). The RMSE (rectangles and solid lines), MBE (triangles and dashed lines) and TSPRD (pluses and dotted lines) are shown. The saw tooth pattern observed is a common feature of such plots in DA. The number of assimilated observations per assimilation time is represented by the green dots.





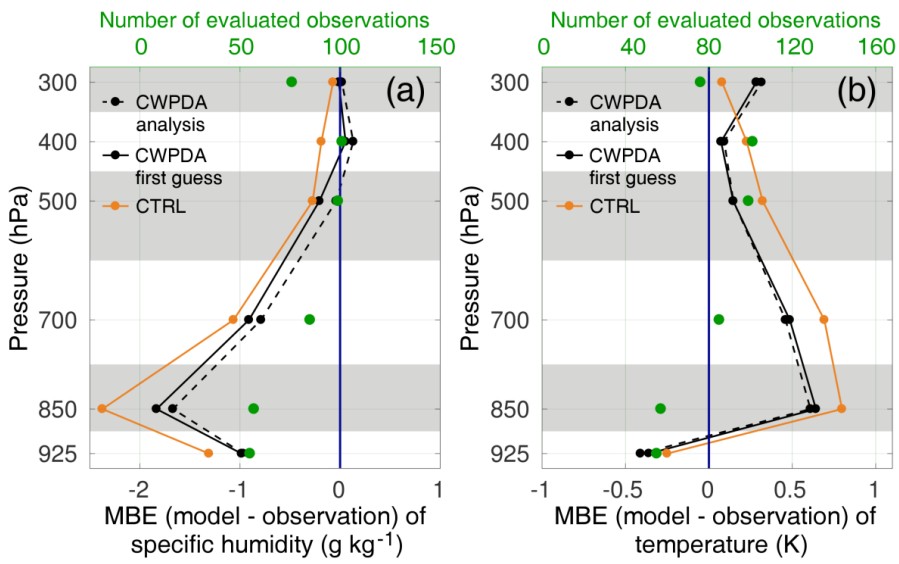

**Figure 5.** Vertical profiles of the MBE per bins of pressure for specific humidity **(a)** and temperature **(b)**, for the two experiments CWPDA (black) and CTRL (orange), with respect to independent radiosonde observations. The solid lines show the ensemble mean first guesses. The dashed line shows the analyses of CWPDA. A dashed line does not exist for CTRL since prior and posterior are identical. The number of evaluated observations per pressure bin is shown in green. All cycling periods listed in Table 1 including 43 radiosoundings at 1200 UTC at Gillot-Aeroport are considered.

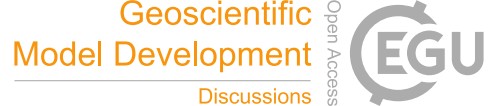

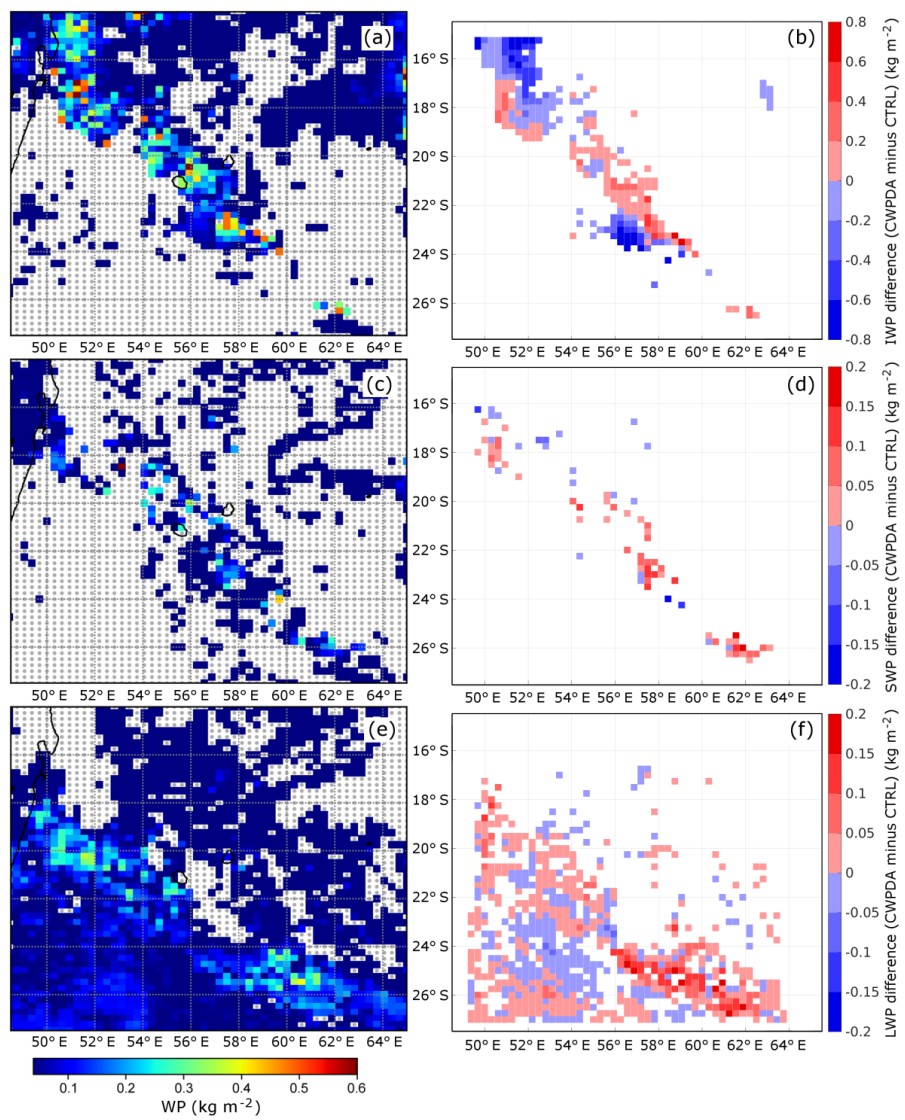

**Figure 6.** Maps of SatCORPS Meteosat-8 gridded CWP retrievals for the three phases IWP **(a)**, SWP **(c)** and LWP **(e)**. Missing observations are shown in grey. The right side shows the difference of the posterior ensemble mean between the two cycling experiments (CWPDA minus CTRL) for the three phases, i.e. $IWP_{diff}$ **(b)**, $SWP_{diff}$ **(d)** and $LWP_{diff}$ **(f)**. Red indicates that CWPDA generates higher WP values than CTRL and blue indicates that CWPDA generates lower values than CTRL. Only locations where the retrievals were successfully assimilated in CWPDA are shown in the right plots. All plots show 12 January 2018, 0000 UTC.



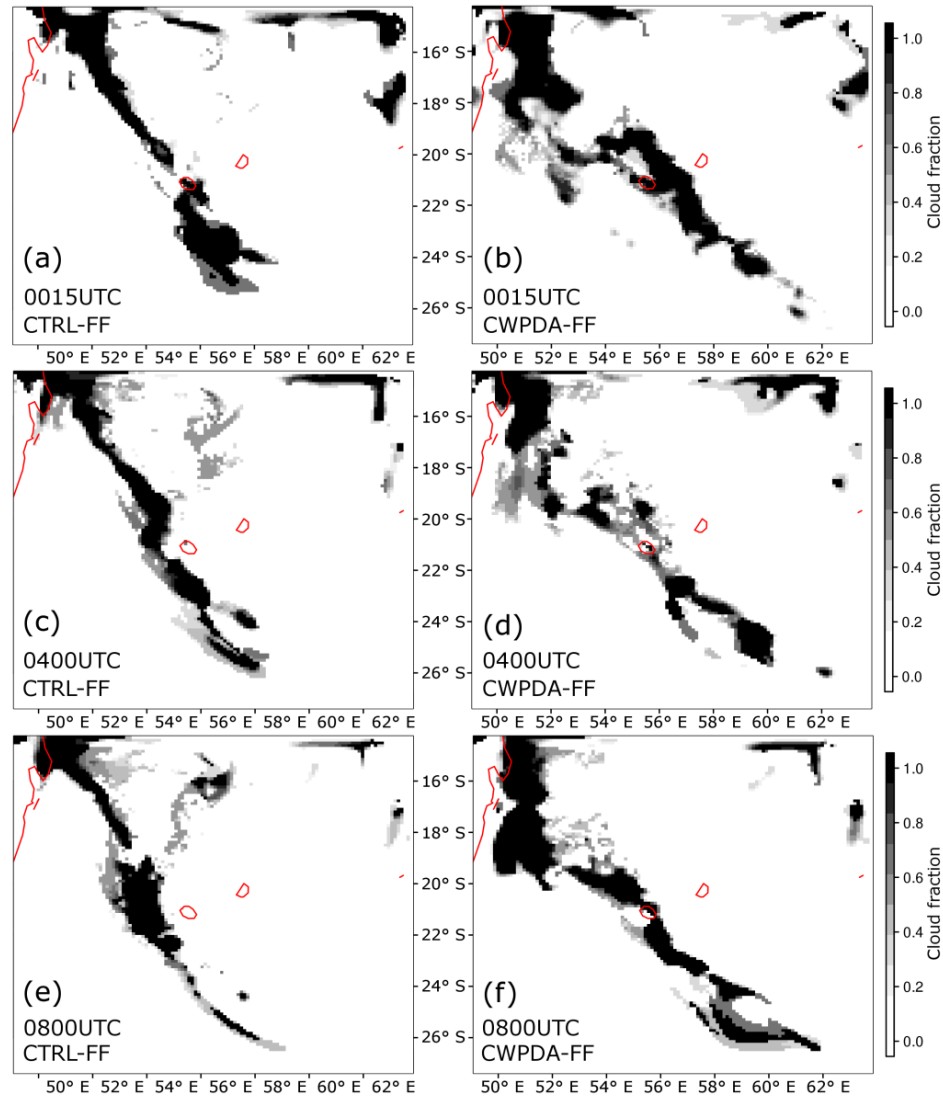

**Figure 7.** WRF forecasts of cloud fraction at 300 hPa from the CTRL-FF **(a, c, e)** and CWPDA-FF **(b, d, f)** experiments on 12 January 2018 at 0015 UTC **(a, b)**, 0400 UTC **(c, d)** and 0800 UTC **(e, f)**. The initial conditions originate from the experiments CTRL and CWPDA respectively. Coastal lines are shown in red.



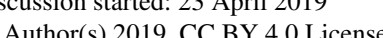



**Figure 8.** GHI on Reunion Island on 12 January 2018 as observed (blue), and forecasted by CWPDA-FF (orange) and CTRL-FF (green) with IDW post-processing for the sites Plaine des Cafres **(a)**, Gros Piron Sainte-Rose **(b)**, Cilaos **(c)**, Bellevue Bras-Panon **(d)**, Le Port **(e)**, Colimacons **(f)**, Piton-Maido **(g)**, Pointe des Trois-Bassins **(h)**, Petite-France **(i)**, Pierrefonds-Aéroport **(j)**, Le Baril **(k)**, Gillot-Aéroport **(l)**.





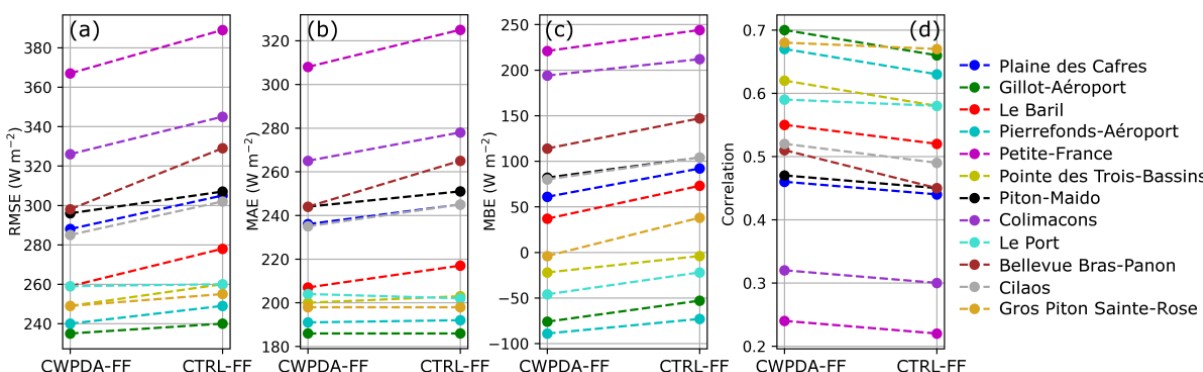

**Figure 9.** WRF intraday irradiance forecast performance at each site in terms of RMSE **(a)**, MAE **(b)**, MBE **(c)** and correlation **(d)**. The difference between the experiments CWPDA-FF and CTRL-FF is shown and IDW has been applied to the GHI forecasts. The free forecasts cover a total of 44 days between 10 December 2017 and 1 March 2018 as listed in Table 1.





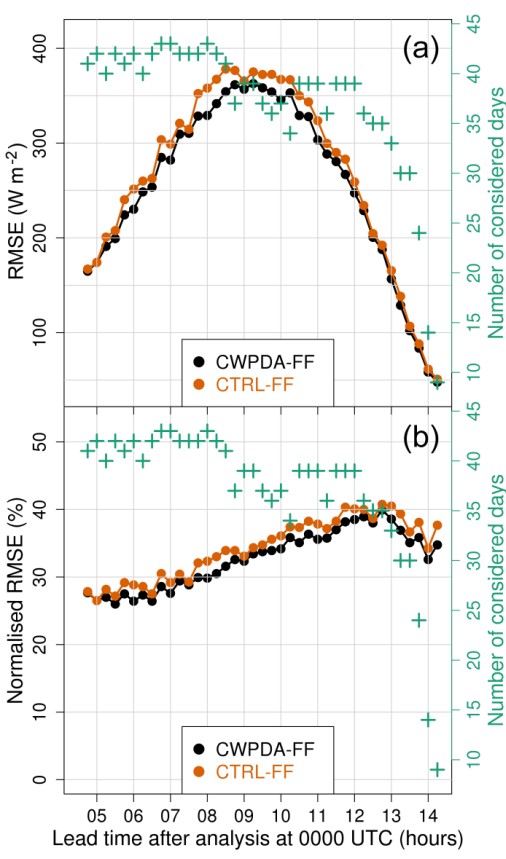

**Figure 10.** Absolute **(a)** and clear-sky normalised **(b)** RMSE of the WRF irradiance forecasts per lead time as a mean over the 12 considered sites on Reunion Island for all free forecast dates listed in Table 1 (44 in total). Values for the two experiments CWPDA-FF (black) and CTRL-FF (orange) are shown. The number of considered days per lead time (green crosses) varies since a given lead time of a given day is considered only when observations are available for all 12 sites. Local time is UTC-4.