# Peer review of "Evaluation of WRF-DART (ARW v.3.9.1.1 and DART manhattan release) multi-phase cloud water path assimilation for short-term solar irradiance forecasting in a tropical environment"

_Geoscientific Model Development, 2019_

## Short Comment (SC1) · 29 Apr 2019

Thank you for specifying exactly which versions of WRF and DART you are using, and pointing to their archival locations. However currently not all the code and data required to run the simulations in the paper is so clearly identified. The input configuration files required to run the simulations are not currently available, and the exact forcing data used in each of those cases is not clear. Please, therefore, archive and reference these configuration files and any pre- or post- processing scripts required to run your experiments or produce the results presented. Note that it is not necessary (and often

not possible) to duplicate all the input data, providing configuration files and specifically identifying input data is sufficient.

The configuration files and scripts should be publicly archived in a persistent archive. Many GMD authors find Zenodo a good choice for this, but other archives are available.

Regards,

David Ham GMD Executive editor.

---

## Referee Comment (RC1) · Anonymous Referee #1 · 10 Jun 2019

[General comments] The data assimilation procedure is consistent with commonly applied DA strategies. The authors did their due diligence in setting up the model configuration, which is typically one of the most challenging aspects of solar forecasting with WRF.

As solar forecasting is challenging on a region-by-region basis due to data availability, topography, and cascading errors from parent NWP models, incremental improvements are important to publish to establish both baselines and reference points for others in similar geographical and climatological regions.

[Figure]

The authors followed solid methodology and provided comprehensive analysis on the results of the simulations.

Overall, I find the quality of the paper to be outstanding and acceptable for publication in its current state.

[Specific comments] I appreciate the background on the large scale flow in the case study, as this information is sometimes not present in studies of this type.

[Technical Corrections] Did not find any.

---

## Referee Comment (RC2) · Anonymous Referee #2 · 20 Jul 2019

The authors investigate the impact of assimilating cloud water path on global horizontal irradiance. The manuscript documents well the background and the originality of the research. The experimental set up is also well described and the explanation of results is clear. The contents of the article should be of interest for GMD readers and I therefore recommend publication of the manuscript. Perhaps the authors can elaborate a bit more about the relatively low improvement in the GHI predictions (3-4%) and potential refinements of their method.

---

## Author Response (AR1)

**Comments of Anonymous Referee #1**

*[General comments] The data assimilation procedure is consistent with commonly applied DA strategies. The authors did their due diligence in setting up the model configuration, which is typically one of the most challenging aspects of solar forecasting with WRF.*

*As solar forecasting is challenging on a region-by-region basis due to data availability, topography, and cascading errors from parent NWP models, incremental improvements are important to publish to establish both baselines and reference points for others in similar geographical and climatological regions.*

*The authors followed solid methodology and provided comprehensive analysis on the results of the simulations.*

*Overall, I find the quality of the paper to be outstanding and acceptable for publication in its current state.*

*[Specific comments] I appreciate the background on the large scale flow in the case study, as this information is sometimes not present in studies of this type.*

*[Technical Corrections] Did not find any.*

**Response to Anonymous Referee #1**

We would like to expressly thank the reviewer for the thorough reading of our manuscript and the acknowledgement of our research.

Kind regards,

Frederik Kurzrock on behalf of all co-authors

**Comments of Anonymous Referee #2**

*The authors investigate the impact of assimilating cloud water path on global horizontal irradiance. The manuscript documents well the background and the originality of the research. The experimental set up is also well described and the explanation of results is clear. The contents of the article should be of interest for GMD readers and I therefore recommend publication of the manuscript. Perhaps the authors can elaborate a bit more about the relatively low improvement in the GHI predictions (3-4%) and potential refinements of their method.*

**Response to Anonymous Referee #2**

Thank you very much for your positive comments on the manuscript.

The stated error reduction of 3-4 % refers to a mean over the 12 considered sites. The site-specific forecast error reduction varies from site to site as a result of Reunion Island's complex topography and diverse microclimates. This circumstance is taken into account in the experiment evaluations in the manuscript, for example in the discussion of results for sites located on the coast or in the mountains. It is true that this aspect should be explained more clearly in the conclusions where we talk about the error reduction of 3-4 %. We therefore propose the following modification at page 15, line 2:

The evaluation of GHI forecasts from the experiments without and with DA shows an overall reduction of 4 % for RMSE and 3 % for MAE due to CWP DA on average over all sites. The method of refining the ICs using CWP DA causes a positive impact on GHI forecasts for the whole duration of the forecast, i.e. up to a lead time of 14 hours. As a consequence of the complex topography of Reunion Island and local thermal circulations, the highest forecast error reduction is achieved for sites located in the mountains and in the lee of the trade winds. Future experiments at higher model resolutions should take this circumstance into account and evaluate the link between the impact of satellite data assimilation and the model's capability to resolve thermally induced local clouds.

Kind regards,

Frederik Kurzrock on behalf of all co-authors

**Comments of the Executive Editor**

*Thank you for specifying exactly which versions of WRF and DART you are using, and pointing to their archival locations. However currently not all the code and data required to run the simulations in the paper is so clearly identified. The input configuration files required to run the simulations are not currently available, and the exact forcing data used in each of those cases is not clear. Please, therefore, archive and reference these configuration files and any pre- or post- processing scripts required to run your experiments or produce the results presented. Note that it is not necessary (and often not possible) to duplicate all the input data, providing configuration files and specifically identifying input data is sufficient.*

*The configuration files and scripts should be publicly archived in a persistent archive. Many GMD authors find Zenodo a good choice for this, but other archives are available.*

*Regards,*

*David Ham GMD Executive editor.*

**Response to Executive Editor**

Dear Executive Editor,

Thank you for your comment and for bringing this issue to our attention.

We have uploaded the namelist files for WRF and DART to Zenodo. Moreover, we will clarify the location of the GFS and GEFS files on the NCEP servers that are required to run WRF.

The following input and configuration files are required to run our simulations:

- The GFS and GEFS input data as specified in the section "Code and data availability".
- The SatCORPS data for Meteosat-8 as specified in the section "Code and data availability".
- The WRF Preprocessing System (WPS) namelist as given in the Zenodo archive.
- The WRF-ARW namelist as given in the Zenodo archive.
- The DART namelist as given in the Zenodo archive.

We have added this information to the paper in the section "Code availability and data availability" on page 15, lines 19-20 and 22-23:

###

The WRF-ARW and DART namelists used in the experiments are available here: https://doi.org/10.5281/zenodo.3354949. The WRF configuration for the cycling and the free forecast experiments is identical.

GFS and GEFS data are available at https://www.ftp.ncep.noaa.gov/data/nccf/com/gfs/prod/ and https://www.ftp.ncep.noaa.gov/data/nccf/com/gens/prod/ respectively.

###

We hope we have solved the issue with this solution.

Kind regards,

Frederik Kurzrock on behalf of all Co-Authors

[revised manuscript text omitted]